# Validation of the Barcelona Magnetic Resonance Imaging Predictive Model for Significant Prostate Cancer Detection in Men Undergoing Mapping per 0.5 Mm-Core Targeted Biopsies of Suspicious Lesions and Perilesional Areas

**DOI:** 10.3390/cancers17030473

**Published:** 2025-01-31

**Authors:** Nahuel Paesano, Violeta Catalá, Larisa Tcholakian, Xavier Alomar, Miguel Ángel Barranco, Jonathan Hernández-Mancera, Berta Miró, Enrique Trilla, Juan Morote

**Affiliations:** 1Department of Surgery, Universitat Autònoma de Barcelona, 08193 Bellaterra, Spain; enrique.trilla@uab.cat; 2Clínica Creu Blanca, 08034 Barcelona, Spain; violetacatala@uroima.com (V.C.); larisa.tcholakian@creublanca.es (L.T.); xavier.alomar@creublanca.es (X.A.); miguelangel.barranco@creublanca.es (M.Á.B.); 3Uroima, 08005 Barcelona, Spain; 4Fundació Puigvert, 08025 Barcelona, Spain; jhernandez@fundacio-puigvert.es; 5Statistics Unit, Vall d’Hebron Research Institute, 08035 Barcelona, Spain; berta.miro@vhir.org; 6Department of Urology, Vall d’Hebron Hospital, 08035 Barcelona, Spain

**Keywords:** prostate biopsy, protocol, significant prostate cancer, detection

## Abstract

The validation of predictive models is essential for informing clinical decisions, particularly in new individual populations or as diagnostic methods advance. This study evaluates the performance of the Barcelona-MRI predictive model (BCN-MRI PM) for identifying significant prostate cancer (sPCa) within the context of an advanced prostate biopsy protocol. In a cohort of 457 men suspected of having PCa, the model demonstrated high accuracy and clinical applicability, reducing unnece-ssary biopsies by 24.9% while maintaining a 95% detection rate for sPCa. These results validate the efficacy of the BCN-MRI PM and support its readiness for clinical implementation in this diagnostic framework.

## 1. Introduction

Prostate cancer (PCa) is the most common malignancy in men and the third leading cause of cancer-specific death worldwide [1]. The European Randomized Screening for Prostate Cancer (ERSPC) trial reported a 20% reduction in PCa-specific mortality among men who underwent screening compared to those in the control arm after seven years of follow-up [2]. Fifteen years later, this reduction has surpassed up to 29%, attributed to the early detection and treatment of significant prostate cancer (sPCa) [3]. Consequently, the focus of early detection of PCa has shifted towards sPCa. This shift has been facilitated by the widespread of prostate-magnetic resonance imaging (MRI), which is now used to select candidates for prostate biopsy and to perform targeted biopsies of suspicious lesions identified through MRI [4]. Despite these advancements, challenges remain, including excessive unnecessary prostate biopsies and the overdetection of insignificant tumors. Current predictive models, based on the Prostate Imaging Reporting and Data System (PI-RADS) and clinical variables, have become essential tools to improve the diagnostic efficiency for sPCa [5]. Since uncertain scenarios remain after the widespread use of MRI, the European Association of Urology recommends the use of risk-stratified pathways for sPCa screening, employing predictive models to guide clinical decision-making [6].

The Barcelona MRI-predictive model (BCN-MRI PM) was developed to assess individual probabilities of sPCa in 1486 men with suspected PCa, excluding those with previous PCa detection, and 5-alpha reductase inhibitors (5-ARI) treatment of symptomatic benign prostatic hyperplasia due to the known modifications of serum PSA levels [7]. All men underwent pre-biopsy multiparametric MRI (mpMRI) reported with the Prostate Imaging-Reporting and Data System (PI-RADS) v 2.0 followed by a 2- to 4-core targeted biopsy of suspicious lesions and/or 12-core systematic biopsies via transrectal route, conducted between 1 January 2016 and 31 December 2019, at one academic institution [8,9]. The predictive variables for sPCa included in this model were age (years), PCa family history (no vs. yes), type of prostate biopsy (initial vs. repeated), serum PSA level (ng/mL), digital rectal examination (normal vs. suspicious), MRI-based prostate volume (mL), and PI-RADS score (1 to 5). A risk calculator was designed being available at the web site https://mripcaprediction.shinyapps.io/MRIPCaPrediction/. The model development was conducted at a single academic institution, and a subsequent initial external validation was conducted in a cohort of 946 men suspected of having PCa who followed the same diagnostic approach employed in the development cohort at two institutions in the me-tropolitan area of Barcelona [10]. The BCN-MRI was the first predictive model for sPCa including the PI-RADS score v 2.0, having no limitation on age, serum PSA level, or prostate volume. The BCN-MRI PM was evaluated for each PI-RADS score, along with the risk calculator incorporating the selection of an appropriate threshold [10,11]. The BCN-MRI PM exhibited higher clinical utility than the prestigious Rotterdam MRI PM in the external validation cohort [12].

Validation of predictive models is critical before their implementation in new populations or when significant changes in the diagnostic approach are introduced to ensure that predictions accurately reflect the true risk of the event occurring [13,14].

Our aim is to validate the BCN-MRI PM in a cohort of men suspected of having PCa, for whom MRI results were assessed with the PI-RADS v 2.1 score, and prostate biopsies were performed using a mapping per 0.5 mm-core targeted biopsy of suspicious lesions and perilesional areas, followed by a 12-core systematic biopsy protocol.

## 2. Materials and Methods

### 2.1. Design, Participants, and Setting

A prospective study was conducted involving 457 men suspected of having PCa who were referred for prostate biopsy to the Creu Blanca reference center in Barcelona, Spain, between 1 February 2022, and 29 February 2024. This study received the approval from the ethics committee of Vall d’Hebron Hospital (PRAG-02/2021).

### 2.2. Diagnostic Approach for sPCa Detection in Prostate Biopsies

Men suspected of having PCa were initially identified based on a serum PSA excee-ding 3.0 ng/mL and/or a suspicious DRE. These individuals underwent external multiparametric MRI (mpMRI) with those exhibiting a PI-RADS score of 3 to 5 referred for transperineal prostate biopsy at Creu Blanca reference center. Additionally, some men with a PI-RADS 2 and a high risk of PCa were also referred for biopsy.

Biparametric MRI (bMRI) was performed at Creu Blanca to segment suspicious lesions using a Siemens Verio™ 3 Tesla MRI scanner (Siemens Healthcare, Erlangen, Germany) equipped with a pelvic-phased array coil. The imaging protocol included T2-weighted transverse sections, along with coronal and sagittal T2-weighted sequences (both 2D and 3D). Diffusion-weighted imaging was acquired with b-values of 1600, 2000, and 3000 s/mm², complemented by apparent diffusion coefficient mapping. An expert uro-radiologist with 15 years of experience and an annual interpretation volume of more than 1000 MRIs, reclassified external mpMRI lesions following the PI-RADS v2.1 criteria [15]. The ProFuse™ system (Eigen, Grass Valley, California, USA) was employed for semi-automatic segmentation of up to three suspicious lesions that satisfied the European Society of Urogenital Radiology standards [16]. Men whose lesions were reclassified to a PI-RADS score of 2 underwent the same biopsy protocol as those with both targeted and systematic biopsies. External mpMRIs were deemed diagnostically adequate, with Prostate Imaging Quality scores of 3 or higher [17,18].

The prostate biopsies were performed via transperineal route under general anesthesia by experienced urologists. Targeted biopsies were obtained every 0.5 mm from suspicious lesions and perilesional areas, followed by a 12-core systematic biopsy using the Artemis^TM^ robotic arm system (Eigen, Gars Valey, CA, USA) and the Aloka Noblus^TM^ ultrasound system (Hitachi, Wilton, CT, USA).

The diagnosis of sPCa was established by an experienced uro-pathologist when the International Society of Urologic Pathology (ISUP) grade group was 2 or higher [19,20].

### 2.3. Assessment of Individual sPCa Likelihood

The endpoint variable of the study was the individual likelihood of sPCa, which was prospectively assessed with the BCN-MRI risk calculator, accessible at the web site https://mripcaprediction.shinyapps.io/MRIPCaPrediction/. The predictive variables in the BCN-MRI risk calculator include age (in years), first-degree family history of PCa (no vs. yes), type of biopsy (initial vs. repeated), serum PSA level (ng/mL), DRE status (normal vs. suspicious), MRI-based prostate volume (mL), and PI-RADS v2.1 scores (from 1 to 5).

### 2.4. Statistical Analysis

Quantitative variables were described as medians with interquartile ranges (IQR), while qualitative variables were expressed as percentages. Descriptive variables were compared with the Mann-Whitney U test and the Chi-square test. Calibration of the predictive model was assessed. Discrimination power of the BCN-MRI PM was determined using the receiver operating characteristic (ROC) curve and the area under the curve (AUC). The net benefit of using the BCN-MRI PM for identifying prostate biopsy candidates was compared to biopsying all men through decision curve analysis (DCA). The clinical utility of the model was determined using the clinical utility curves (CUC), which explored the potential rates of missed sPCa detection and the avoidable prostate biopsies. Specificities at sensitivity threshold of 100%, 97.5%, 95%, 92.5%, and 90% were analyzed. The efficacies analyzed with the Youden index, and clinically the rates of saved prostate biopsies and undetected sPCa were analyzed. Transparent reporting of a multivariable prediction model for individual prognosis or diagnosis (TRIPOD) statements were followed [21]. Statistical analyses were computed using R programming language v.4.0.3 (The R Foundation for Statistical Computing, Vienna, Austria) and SPSS v.24 (IBM, statistical package for social sciences, San Francisco, CA, USA).

## 3. Results

### 3.1. Characteristics of the Validation Cohort

The most relevant characteristics of the validation cohort, consisting of 457 men suspected of having PCa, are presented in Table 1. The age of participants ranged from 32 to 88 years, with 151 being older than 70 (33%) and 3 (0.6%) younger than 45. Serum PSA levels were below 3.0 ng/mL in one case (2.8 ng/mL) and above 10 ng/mL in 82 cases (17.9%). The MRI-derived prostate volume was up to 50 mL in 260 men (56.9%), between 51 and 100 mL in 152 cases (33.7%), and above 100 mL in 45 (9.8%).

There was one suspicious lesion in 360 cases (78.8%), two lesions in 93 (20.4%) and three lesions in four cases (0.9%). The median size of the index lesion was 12 mm (IQR 4–44). In 298 cases (65.2%), the index lesion was in the peripheral zone of the prostate gland; in 148 cases (32.4%), it was in the central/transitional zone; and in 11 cases (2.4%), it was in the fibromuscular anterior zone. The median overall number of core-obtention per case was 21 (IQR 19–24) ranging from 14 to 35. The overall detection rate of sPCa was 58.4%, and those according to the PI-RADS category of the index lesion were 2.7% in PI-RADS 2, 33% in PI-RADS 3, 80% in PI-RADS 4, and 87.4% in PI-RADS 5.

### 3.2. Calibration of the BCN-MRI PM in the Validation Cohort

Calibration of the BCN-MRI PM in this validation cohort of 457 men suspected of having PCa, in whom 267 sPCa (58.4%) were detected, is presented in Figure 1. The calibration curve closely aligns with the ideal curve, with a slight over detection of sPCa at the median threshold points. Calibration curves obtained in the development cohort of the BCN-MRI PM (B), and the initial external validation cohort (C) are also shown.

### 3.3. Discrimination Ability, Net Benefit, and Clinical Utility of the BCN-MRI PM for sPCa Detection

The discriminatory ability for sPCa of the BCN-MRI PM, represented with ROC curves, showed an AUC of 0.862 (95% CI 0.828–0.896) in this validation cohort, Figure 2A. This result was comparable with the AUC of 0.842 (95% CI 0.822–0.861) observed in the development cohort of the BCN-MRI PM, and better than 0.743 (95% CI: 0.711–0.776) observed in the initial validation cohort (*p* = 0.008), Figure 2B [10].

The DCA showed a net benefit of the BCN-MRI PM over biopsy for all men, beginning at a threshold probability of 12%, Figure 3A. This benefit was comparable with that observed in the development cohort and the initial validation cohort, Figure 3B [10].

The clinical utility of the BCN-MRI PM, based on the estimation of the percentages of saved biopsies and undetected sPCa for 5% threshold intervals, is presented is presented in Table 2.

Table 3 displays the specificities and 95% confidence intervals of the BCN-MRI PM at sensitivities of 100%, 97.5%, 95%, 92.5%, and 90% for sPCa, as these represent the most relevant clinical scenarios. The efficacy measured by the Youden index, and the corresponding percentage of avoided biopsies are also presented. If we choose to allow up to 5% of sPCa to go undetected, the specificity of the BCN-MRI PM will be 48.4% (95% CI, 42.3–51.6), the Youden index will be 43.6, and 24.9% of prostate biopsies will be avoided.

Clinical utility curves (CUCs), representing the differential percentage between undetected sPCa and saved prostate biopsies across a continuous evolution of the threshold probability point from 0 to 100%, are shown in Figure 4. A highly relevant scenario, based on the detection of at least 95% of sPCa, is highlighted with arrows corresponding to the application of the 12% threshold of the BCN-MRI PM, which saves almost 25% of prostate biopsies.

### 3.4. Potential Clinical Utility of the BCN-MRI PM According to the PI-RADS v 2.1 Score

Analyzing the potential clinical utility of the BCN-MRI PM in individuals with PI-RADS 2, the probability of sPCa ranged from 0.22% to 26.70% among the 73 cases analyzed. The probabilities in the two cases with significant PCa (sPCa) detected were 11.28% and 22.99%, with respective PSA densities of 0.14 and 0.17, and normal digital rectal examinations in both. By applying an 11% threshold for recommending biopsy in PI-RADS 2 cases, a 100% sensitivity for sPCa cases would be detected, while prostate biopsies using this threshold would be indicated in 16 cases (21.9%) within this subgroup.

For PI-RADS 3 (the most uncertain scenario), 35 cases of sPCa were identified among 106 individuals. By applying a 10% threshold, 36 prostate biopsies (40%) could be avoided while 8 sPCa cases (22.8%) would be missed, representing a global loss of 3% among the 267 significant PCa cases diagnosed in the overall series.

For PI-RADS 4 and 5, no threshold was founded for allowing to avoid prostate biopsies in theese subsets without missing sPCa cases.

## 4. Discussion

The BCN-MRI PM for sPCa has been successfully validated in men suspected of ha-ving PCa through a highly effective transperineal prostate biopsy protocol. This approach involved obtaining core obtention cores at 0.5 mm intervals from suspicious lesions and perilesional areas, along with a 12-core systematic biopsy [22,23,24]. Among individuals with a PI-RADS score 2, sPCa was detected in 2.7% of cases, a rate lower than the 9% (95% CI 5–13) reported in the meta-analysis by Oerther et al. on sPCa detection rates across PI-RADS v2.1 assessment categories. Conversely, a detection rate of 33% was observed in men with a PI-RADS score of 3, which is notably higher than the referenced rate of 16% (95% CI: 7–27). For those with a PI-RADS score of 4, this approach identified sPCa in 80% of cases, surpassing the 59% (95% CI: 39–78) reported in the meta-analysis. Finally, in men with a PI-RADS score of 5, the protocol achieved an sPCa detection rate of 87.4%, placing it at the upper end of the range reported in the meta-analysis, which cited a rate of 85% (95% CI: 73–94) [25].

The clinical characteristics of the analyzed cohort differed from those of the deve-lopment cohort and the initial external validation cohort of the BCN-MRI PM in several aspects, including age, serum PSA level, prostate volume, percentage of suspicious DRE findings, proportion of biopsy naïve men, family history of PCa, and the distribution of PI-RADS scores. Notably, the most significant difference was observed in the outcome variable of the BCN-MRI PM. The rate of sPCa detected in the present validation cohort was 58.3% compared to 36.9% in the development cohort and 40.8% in the initial external validation cohort. The latter utilized PI-RADS v2.0 and employed a biopsy protocol consisting of 2-to 4-core targeted biopsies of suspicious lesions alongside a 12-core systematic biopsy performed via transrectal route [10]. Despite these differences, the performance of the BCN-MRI PM under these challenging conditions was comparable to its performance in the initial external validation. The model demonstrated discrimination abilities with an AUC of 0.862 (IQR 0.828–0.896) and 0.858 (IQR 0.833–0.883), respectively. In both validation cohorts, applying the BCN-MRI PM provided a net benefit compared to perfor-ming biopsies on all men, beginning at a threshold of approximately 12%. Furthermore, the clinical utility of the BCN-MRI PM, as indicated by the percentage of saved prostate biopsies at 95% sensitivity, was 24.9% in the present cohort compared to 23.7% in the initial external validation. Thus, we conclude that the BCN-MRI PM performed slightly better in the current cohort, where enhanced biopsy effectiveness was observed compared to the protocol used in the development and initial external validation cohorts [10].

In recent years, several significant changes have been introduced to the diagnostic pathway for sPCa, potentially influencing the effectiveness of prostate biopsy. The currently utilized PI-RADS version 2.1 differs from version 2.0, which was employed in the development and initial validation cohort of the BCN-MRI PM [9,16]. Additionally, the currently recommended approach for performing prostate biopsies is the transperineal route, whereas the transrectal route was used during the development and initial external validation of the BCN-MRI PM [26]. These updates underscore the need to validate the BCN-MRI PM under these revised conditions. Since the development an initial external validation excluded men receiving 5-alpha reductase inhibitors (5-ARI) for symptomatic benign prostatic hyperplasia due to their impact on serum PSA levels, the BCN-MRI PM has now been validated in a cohort of men suspected of having PCa who are undergoing 5-ARI treatment [27,28]. Furthermore, the predictive model has been validated in contexts where version 2.1 of the PI-RADS score is used in MRI reports and prostate biopsies are performed via transperineal route [29]. Lastly, given that an optimal prostate biopsy protocol has not yet been defined, and evidence suggests that mapping biopsies of targeted suspicious lesions and perilesional areas may enhance biopsy effectiveness [29,30], we validated the BCN-MRI PM in a cohort of men undergoing this biopsy protocol.

The BCN-MRI PM has been successfully validated in men suspected of having PCa who underwent targeted biopsy of lesional and perilesional following a 0.5 mm-core mapping protocol in combination with a 12-core systematic biopsy. Its potential clinical utility of the BCN-MRI PM was specifically evaluated across different PI-RADS v2.1 categories. We found that in individuals with PI-RADS 2, using a 11% threshold, it could detect 100% of existing clinically significant prostate cancers (sPCa) in this group by biopsying only 22%. Similarly, for PI-RADS 3, using a 10% threshold, it could potentially avoid 40% of prostate biopsies, although this would result in missing 23% of detected sPCa, representing a 3% reduction in the overall sPCa detection rate in this series. For PI-RADS 4 and 5, no threshold was identified that would allow avoiding biopsies without missing some sPCa cases.

Several limitations can be attributed to this study. Performing biopsies on men reclassified as PI-RADS core 2 may introduce a bias into the detection rate of sPCa. Nevertheless, this approach may also be considered a strength, as it provides an opportunity to assess the effectiveness of applying the BCN-MRI PM in a subset of men who are rarely subjected to prostate biopsy. A validation conducted in a cohort referred from a non-structured, opportunistic screening effort for early sPCa detection does not necessarily guarantee successful validation in the contexts of a population-based screening program. Additionally, it was not possible to assess the specific number and contribution of perilesional cores in detecting sPCa, as cores obtained from each suspicious lesion and its perilesional area were collectively analyzed. A limitation of the BCN-MRI PM lies in its reliance on the definition of sPCa based on findings from prostate biopsies, which may not fully represent the true pathology of the entire prostate gland. It is important to acknowledge the inherent constraints associated with predictive models. These models estimate individual probabilities of a condition based on the characteristics on the development cohort. Since PI-RADS score is incorporated in MRI PMs, its potential utility should be analyzed in each category to identify the scenarios in which they are effective [10,31]. We have observed that the BCN-MRI PM has potential utility to select candidates for prostate biopsy with PI-RADS 2 and 3. However, in PI-RADS 4 and 5, in which sPCa detection was very high, no potential saving biopsies was observed without missing sPCa.

The sequenced application of the BCN PMs (BCN-1 PM prior MRI and BCN-MRI PM) for the early detection sPCa represents a cost-effective approach. This strategy is advantageous when compared to the recommended practice of performing MRI for all men suspected of having PCa and conducting biopsies on those with PI-RADS 3 or higher, as well with the systematic use of novel tumor markers [32].

Predictive models face challenges due to the ongoing changes in the characteristics of the populations where they are implemented, often requiring recalibration and readjustment to maintain accuracy [30]. Achieving real-time updates for predictive models remains a significant challenge for the future risk calculators [33]. Establishing a continuous feedback loop incorporating new cases, leveraging appropriate machine learning algorithms, and utilizing federated networks risk calculators to undergo continuous updates at each participating site, thereby ensuring sustained accuracy [34]. Moreover, the broader applications of novel imaging techniques and predictive modelling in cancer detection are expected to contribute to the future prediction of sPCa [35,36,37,38,39,40,41,42,43,44,45,46,47].

## 5. Conclusions

The BCN-MRI PM has been successfully validated in men suspected of having PCa using an effective prostate biopsy protocol that includes 0.5 mm mapping-core targeted biopsies of suspicious lesions and 12-core systematic biopsy.

## Figures and Tables

**Figure 1 cancers-17-00473-f001:**
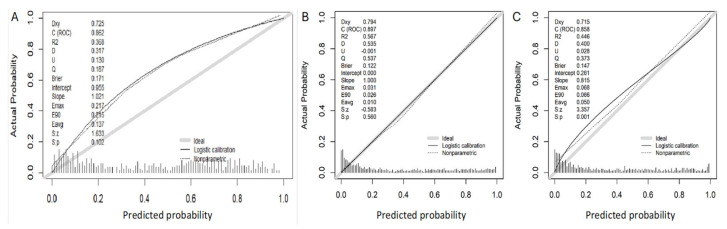
Calibration curve of the BCN-MRI predictive model in the present validation cohort (**A**), in the development cohort (**B**), and in the initial external validation cohort (**C**).

**Figure 2 cancers-17-00473-f002:**
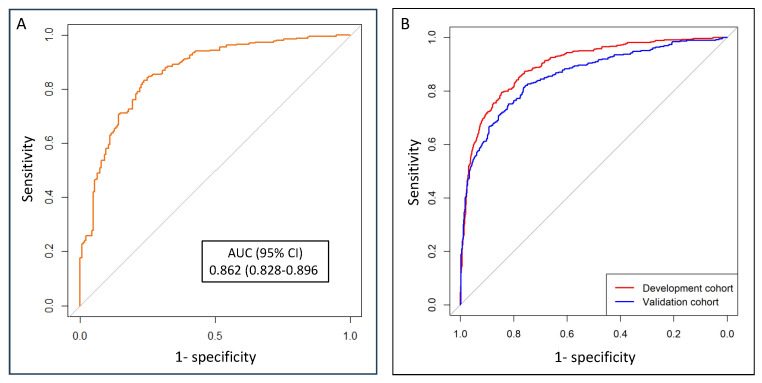
Discrimination ability of the BCN-MRI predictive model for significant PCa detection in prostate biopsy in this validation cohort (**A**), referenced to those observed in the development cohort and the initial external validation cohort (**B**) [10].

**Figure 3 cancers-17-00473-f003:**
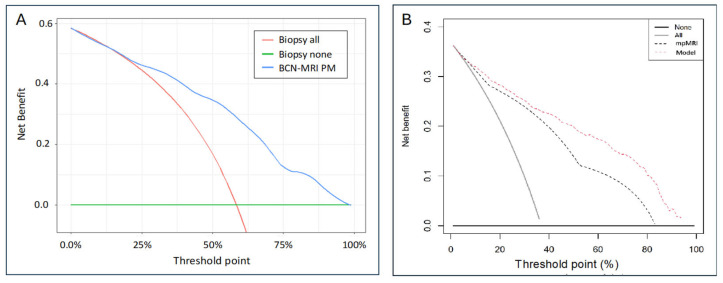
Net benefit of the BCN-MRI predictive model, represented trough Decision Curve Ana-lysis (DCA), in which ‘y axis’ represents to biopsy all men and ‘x axis’ represents to biopsy any of them. The DCA showed a net benefit of the BCN-MRI PM over biopsy for all men, be-ginning at a threshold probability of 12%, (**A**). This benefit was comparable with that observed in the development cohort and the initial validation cohort, (**B**) [10].

**Figure 4 cancers-17-00473-f004:**
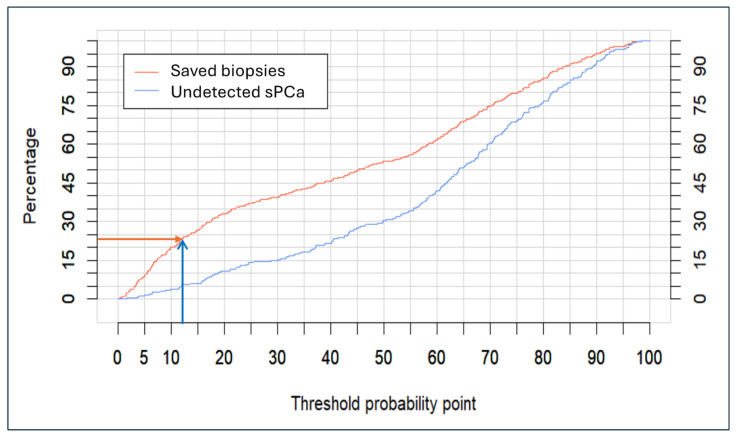
Analysis of clinical utility of the BCN-MRI predictive model, represented by clinical utility curves (CUCs) showing the undetected rate of significant PCa and avoided prostate biopsies according to the continuous threshold point. Arrows indicate the rate of saved biopsies at the 12% threshold corresponding to the 95% sensitivity for significant PCa detection.

**Table 1 cancers-17-00473-t001:** Characteristics of the validation cohort.

Characteristic	Measure
Number of men	457
Median age, years (IQR)	67 (56–72)
Median serum PSA, ng/mL (IQR)	6.0 (4.8–9.0)
Suspicious DRE, *n* (%)	61 (13.3)
PCa family history, *n* (%)	80 (17.5)
Repeated prostate biopsies, *n* (%)	107 (23.4)
Median prostate volume, ml (IQR)	47 (34–63)
PI-RADS score 2, *n* (%)	73 (16.0)
PI-RADS score 3, *n* (%)	106 (23.2)
PI-RADS score 4, *n* (%)	175 (38.3)
PI-RADS score 5, *n* (%)	103 (22.5)
Significant PCa detection, *n* (%)	267 (58.4)

IRQ: 25–75 percentiles; *n*: number; PSA: prostate-specific antigen; DRE: digital rectal examination; PCa: prostate cancer; PI-RADS: prostate imaging-reporting and data system.

**Table 2 cancers-17-00473-t002:** Clinical efficacy of the BCN-MRI predictive model showing the rate of undetected sPCa and saved biopsies to each 5% threshold from 0 to 100 percent.

Threshold (%)	Undetected sPCa (%)	Saved Biopsies (%)
0	0.0	0.0
5	1.5	9.2
10	3.7	20.1
15	6.0	26.7
20	10.9	33.0
25	14.2	37.2
30	15.0	39.4
35	18.4	42.7
40	21.7	45.7
45	27.3	49.9
50	30.3	53.4
55	34.1	56.0
60	41.9	61.7
65	50.9	68.7
70	60.3	74.8
75	68.9	79.9
80	76.8	85.8
85	84.3	90.8
90	91.4	95.0
95	96.6	98.0
100	100.0	100.0

**Table 3 cancers-17-00473-t003:** Specificities estimated from 90 to 100% sensitivities of the BCN-MRI predictive model, efficacy of the model ‘Youden index’ and saved biopsies with these thresholds.

Sensitivity (%)	Threshold (%)	Specificity (%) (95% CI)	Youden Index	Saved Biopsies (%)
100	1.6	5.3 (3.8–6.1)	5.3	1.6
97.5	7.8	34.3 (30.4–40.2	25.8	17.1
95	11.9	48.4 (42.3–51.6)	43.6	24.9
92.5	16.5	59.5 (53.4–63.5)	52.0	30.2
90	18.7	62.6 (59.7–65.7)	52.9	32.6

## Data Availability

Database is available under request to corresponding author.

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
