# Peer review of "Validation of the Barcelona Magnetic Resonance Imaging Predictive Model for Significant Prostate Cancer Detection in Men Undergoing Mapping per 0.5 Mm-Core Targeted Biopsies of Suspicious Lesions and Perilesional Areas"

_cancers, 2025, doi:10.3390/cancers17030473_

Round 1

Reviewer 1 Report

Comments and Suggestions for Authors

The manuscript aims to validate the Barcelona-MRI predictive model (BCN-MRI PM) for detecting significant prostate cancer (sPCa) using a new biopsy protocol. In a prospective cohort of 457 men, the model demonstrated high accuracy and clinical utility, reducing unnecessary biopsies by 24.9% while maintaining 95% sensitivity for sPCa detection.

The study's aim is clear: optimizing prostate cancer detection while minimizing unnecessary biopsies. The validation design is robust, with prospective data collection and application of a rigorous biopsy protocol. The study aligns well with current advancements in prostate cancer diagnostics, particularly the shift towards MRI-guided biopsies and risk-stratified approaches.

However, there are several aspects that need to be addressed:

First, including PI-RADS 2 lesions in the biopsy protocol raises potential methodological and ethical concerns. These patients, typically with a low likelihood of harboring significant prostate cancer (sPCa), may not yield substantial diagnostic benefits, yet their inclusion could affect the study outcomes in several ways, such as specificity and net benefit. The low diagnostic yield of biopsies in this group could potentially skew the validation results of the BCN-MRI PM. Excluding these low-risk patients could improve the precision and overall applicability of the model in clinical practice. A sensitivity analysis that excludes PI-RADS 2 patients could provide valuable insights and enhance the robustness of the findings.

Second, biopsying patients with PI-RADS 2 lesions introduces ethical questions regarding risk versus benefit. Given the low probability of detecting sPCa, how is the decision to biopsy such lesions justified within the study protocol? Additionally, how are patients informed about the rationale for biopsying PI-RADS 2 lesions, and what measures are taken to ensure that they understand the limited diagnostic benefit? Clarifying this in the manuscript is essential to address potential ethical implications.

Third, the study's reliance on biopsy-proven sPCa as the endpoint variable may not fully reflect the true pathology of the prostate gland. This limitation has significant implications for the model's validation. Without data from radical prostatectomy specimens or other comprehensive histological findings, the model cannot account for undetected or misclassified sPCa. This constraint reduces the algorithm's predictive strength and clinical reliability. The authors should expand on this limitation in the discussion.

Finally, the study does not provide subgroup analyses that could shed light on the model's performance in distinct clinical scenarios. This omission limits the ability to assess the generalizability and utility of the BCN-MRI PM across different patient populations. For example, the model's calibration and discrimination may differ significantly between biopsy-naïve patients and those undergoing repeat biopsies, or across various age groups. Including subgroup analyses would help clarify these variations and improve the model's clinical applicability in diverse settings.

Author Response

Responses to Reviewer 1

We appreciate the Reviewer’s comments and suggestions, as well as the opportunity to make revisions to the text that improve the manuscript. The changes made to the manuscript have been highlighted in red.

Comments and Suggestions for Authors

The manuscript aims to validate the Barcelona-MRI predictive model (BCN-MRI PM) for detecting significant prostate cancer (sPCa) using a new biopsy protocol. In a prospective cohort of 457 men, the model demonstrated high accuracy and clinical utility, reducing unnecessary biopsies by 24.9% while maintaining 95% sensitivity for sPCa detection. The study's aim is clear: optimizing prostate cancer detection while minimizing unnecessary biopsies. The validation design is robust, with prospective data collection and application of a rigorous biopsy protocol. The study aligns well with current advancements in prostate cancer diagnostics, particularly the shift towards MRI-guided biopsies and risk-stratified approaches.

However, there are several aspects that need to be addressed:

  1. Including PI-RADS 2 lesions in the biopsy protocol raises potential methodological and ethical concerns. These patients, typically with a low likelihood of harboring significant prostate cancer (sPCa), may not yield substantial diagnostic benefits, yet their inclusion could affect the study outcomes in several ways, such as specificity and net benefit. The low diagnostic yield of biopsies in this group could potentially skew the validation results of the BCN-MRI PM. Excluding these low-risk patients could improve the precision and overall applicability of the model in clinical practice. A sensitivity analysis that excludes PI-RADS 2 patients could provide valuable insights and enhance the robustness of the findings.

Response: Thank you for this insightful comment. Indeed, current prostate cancer (PCa) guidelines recommend against performing biopsies in men with suspected PCa who have a PI-RADS score of <3, unless they exhibit an elevated risk of significant PCa (sPCa). According to version 2.1, the probability of detecting sPCa detection in men with PI-RADS 2 score is 9% (95%CI 5-13) (Oerther et al. Prostate Cancer Prostatic Dis. 2022;25,256-263.) This highlights a degree of uncertainty when avoiding biopsies in PI-RADS 2 lesions, as it is possible to miss 5 and 13% of sPCa cases, which presents also ethical concerns. Our study aimed to validate the BCN-MRI predictive model (PM) using a biopsy scheme different from that employed in the development cohort. The BCN-MRI PM integrates all PI-RADS scores, and we believe that including PI-RADS 2 scores enhances the model's validation by potentially identifying cases where biopsy would detect sPCa. The key aspect of a validation study is to achieve the highest overall sensitivity for detecting sPCa, while minimizing the number of prostate biopsies, Additionally, it is possible to evaluate the model's performance across different PI-RADS categories, as we demonstrated in the development cohort (Morote et al., Cancers 2022; 14,1589).

  1. Biopsying patients with PI-RADS 2 lesions introduces ethical questions regarding risk versus benefit. Given the low probability of detecting sPCa, how is the decision to biopsy such lesions justified within the study protocol? Additionally, how are patients informed about the rationale for biopsying PI-RADS 2 lesions, and what measures are taken to ensure that they understand the limited diagnostic benefit? Clarifying this in the manuscript is essential to address potential ethical implications.

Response: Thank you for this comment. In this study, most men referred for biopsy had a PI-RADS score of 3 or higher, based on external multiparametric MRI (mpMRI). At the reference center, biparametric MRI is routinely performed to segment lesions, accompanied by an expert evaluation of PI-RADS v2.1. In this context, 73 men (16%) were reclassified to a PI-RADS v2.1 score of 2. These men underwent biopsy at the discretion of the physician referring the patient with the external mpMRI (lines 104–122). The clinical benefits of expert interpretation are comparable to those previously reported by our group (Paesano et al., Clinical Genitourinary Cancer, 2024; 22: 102233).

  1. The study's reliance on biopsy-proven sPCa as the endpoint variable may not fully reflect the true pathology of the prostate gland. This limitation has significant implications for the model's validation. Without data from radical prostatectomy specimens or other comprehensive histological findings, the model cannot account for undetected or misclassified sPCa. This constraint reduces the algorithm's predictive strength and clinical reliability. The authors should expand on this limitation in the discussion.

Response: Thank you again. The BCN-MRI predictive model was developed to estimate the probability of sPCa in prostate biopsy results. Therefore, any validation must be conducted based on this premise. It is true that biopsy results are not fully representative of the entire prostate gland and can only be verified in patients undergoing radical prostatectomy. This limitation constrains the predictive strength of the model, which must be acknowledged, even though, in practice, the outcome variable is the biopsy result. This limitation is noted in lines 313–315.

  1. The study does not provide subgroup analyses that could shed light on the model's performance in distinct clinical scenarios. This omission limits the ability to assess the generalizability and utility of the BCN-MRI PM across different patient populations. For example, the model's calibration and discrimination may differ significantly between biopsy-naïve patients and those undergoing repeat biopsies, or across various age groups. Including subgroup analyses would help clarify these variations and improve the model's clinical applicability in diverse settings.

Response: We agree with the arguments presented by the Reviewer. However, the BCN-MRI predictive model was developed using the identified independent predictive variables. This means that any validation must include the model's variables within the population group being analyzed. This does not diminish the importance of examining the model's validity in specific subgroups, which remains of great interest. Specifically, the BCN-MRI predictive model was analyzed based on PI-RADS categories, with the conclusion that its benefit in PI-RADS 5 is limited, as biopsies cannot be avoided without missing sPCa detections (Morote et al., Cancers, 2022;14:1589). The decision to perform a biopsy is always a clinical one, based on the probability of sPCa estimated by the predictive model and the characteristics of the man with suspected PCa.

Reviewer 2 Report

Comments and Suggestions for Authors

1: The introduction could provide better context on what the BCN-MRI PM is where it came from, and why it is different from any other model.

2:Explain the approach; more details for suspicious lesions identification and characterization, and the methodology of PI-RADS v2. Incorporation of 1 Scores into Predictive Model

3: While if we look for AUC results they are on point, however comparison with other models or clinical standard output for better insight of the results would be welcome.

4: Emphasize sensitivity, specificity, and confidence intervals in Results much more and provide estimator-based results in Results instead of Interpretations and Discussion sections only to present the results more robustly.

5: Strong clinical utility section, but a broader impact section discussing patient outcomes or cost-effectiveness might improve relevance.

6: Transparency in experimental methods and results appears to be a strength of this study, however, reporting of the generalizability of the model results across differing patient populations may be required to improve the paper further.

7: Further explanation of BCN-MRI risk calculator usability and integration into clinical systems would be helpful.

8: Providing graphical representations of the model performance, like ROC curves or calibration plots, could help minimize this problem.

9:Reduce the similarity, as it is copied from two papers, and the author has to make an effort to reduce the plagiarism.

10: References can be cited that will help to better understand the broader applications of novel imaging techniques and predictive modelling in cancer detection and treatment field Papers like 

  1. Wang, H., Yan, Z., Yang, W., Liu, R., Fan, G., Gu, Z.,... Tang, Z. (2025). A strategy of monitoring acetylcholinesterase and screening of natural inhibitors from Uncaria for Alzheimer's disease therapy based on near-infrared fluorescence probe. Sensors and Actuators B: Chemical, 424, 136895. doi: https://doi.org/10.1016/j.snb.2024.136895
  1. He, Y., Bao, M., Chen, Y., Ye, H., Fan, J.,... Shi, G. (2024). Accuracy characterization of Shack–Hartmann sensor with residual error removal in spherical wavefront calibration. Light: Advanced Manufacturing, 4(4), 393-403. doi: 10.37188/lam.2023.036
  1. Xu, X., Luo, Q., Wang, J., Song, Y., Ye, H., Zhang, X.,... Shi, G. (2024). Large-field objective lens for multi-wavelength microscopy at mesoscale and submicron resolution. Opto-Electronic Advances, 7(6), 230212. doi: 10.29026/oea.2024.230212
  1. Zou, Y., Zhu, S., Kong, Y., Feng, C., Wang, R., Lei, L.,... Chen, L. (2024). Precision matters: the value of PET/CT and PET/MRI in the clinical management of cervical cancer. Strahlentherapie und Onkologie. doi: https://doi.org/10.1007/s00066-024-02294-8
  1. Yao, X., Zhu, Y., Huang, Z., Wang, Y., Cong, S., Wan, L.,... Hu, Z. (2024). Fusion of shallow and deep features from 18F-FDG PET/CT for predicting EGFR-sensitizing mutations in non-small cell lung cancer. Quantitative Imaging in Medicine and Surgery 2024, 14(8), 5460-5472. doi: 10.21037/qims-23-1028
  1. Sun, T., Lv, J., Zhao, X., Li, W., Zhang, Z.,... Nie, L. (2023). In vivo liver function reserve assessments in alcoholic liver disease by scalable photoacoustic imaging. Photoacoustics, 34, 100569. doi: https://doi.org/10.1016/j.pacs.2023.100569
  1. Du, Y., Chen, L., Yan, M., Wang, Y., Zhong, X., Xv, C.,... Cheng, Y. (2023). Neurometabolite levels in the brains of patients with autism spectrum disorders: A meta-analysis of proton magnetic resonance spectroscopy studies (N = 1501). Molecular Psychiatry, 28(7), 3092-3103. doi: 10.1038/s41380-023-02079-y
  1. Xiang, Y., Jialing, W., Jianhao, L., Jiangpeng, A., Feizhou, D.,... Rui, J. (2024). Bi-Parametric Magnetic Resonance Imaging Analysis of Biochemical Recurrence of Prostate Cancer after Radical Surgery and Its Predictive Value: A Retrospective Study. Archivos Españoles de Urología, 77(5), 598-604. doi: 10.56434/j.arch.esp.urol.20247705.81
  1. Lan, Z., Tan, F., He, J., Liu, J., Lu, M., Hu, Z.,... Huang, Y. (2024). Curcumin-primed olfactory mucosa-derived mesenchymal stem cells mitigate cerebral ischemia/reperfusion injury-induced neuronal PANoptosis by modulating microglial polarization. Phytomedicine, 129, 155635. doi: https://doi.org/10.1016/j.phymed.2024.155635
  1. Li, W., Wu, J., Zhang, J., Wang, J., Xiang, D., Luo, S.,... Liu, X. (2018). Puerarin-loaded PEG-PE micelles with enhanced anti-apoptotic effect and better pharmacokinetic profile. Drug Delivery, 25(1), 827-837. doi: 10.1080/10717544.2018.1455763
  1. Gao, X., Tang, J., Liu, H., Liu, L., & Liu, Y. (2019). Structure–activity study of fluorine or chlorine-substituted cinnamic acid derivatives with tertiary amine side chain in acetylcholinesterase and butyrylcholinesterase inhibition. Drug Development Research, 80(4), 438-445. doi: https://doi.org/10.1002/ddr.21515
  1. Lu, Q., Chen, Y., Liu, H., Yan, J., Cui, P., Zhang, Q.,... Liu, Y. (2020). Nitrogen-containing flavonoid and their analogs with diverse B-ring in acetylcholinesterase and butyrylcholinesterase inhibition. Drug Development Research, 81(8), 1037-1047. doi: https://doi.org/10.1002/ddr.21726

  1. Wang, Y., Xu, Y., Song, J., Liu, X., Liu, S., Yang, N.,... Zhang, Y. (2024). Tumor Cell-Targeting and Tumor Microenvironment–Responsive Nanoplatforms for the Multimodal Imaging-Guided Photodynamic/Photothermal/Chemodynamic Treatment of Cervical Cancer. International Journal of Nanomedicine, 19, 5837-5858. doi: 10.2147/IJN.S466042
Comments on the Quality of English Language

The English could be improved to more clearly express the research.

Author Response

Responses to Reviewer 2

We appreciate the Reviewer’s comments and suggestions, as well as the opportunity to make revisions to the text that improve the manuscript. The changes made to the manuscript have been highlighted in red.

 Comments and Suggestions for Authors

  1. The introduction could provide better context on what the BCN-MRI PM is where it came from, and why it is different from any other model.

Response: Thank you. The details of the development and initial external validation of the BCN-MRI predictive model (PM) are specified in lines 65-86 of the introduction. However, we have expanded on its distinguishing features compared to other developed models, based on our own systematic review of the literature (Triquell et al.), with the added reference [11].

  1. Explain the approach; more details for suspicious lesions identification and characterization, and the methodology of PI-RADS v2. Incorporation of 1 Scores into Predictive Model.

Response: Thank you. The details of the external mpMRIs with which participants were referred to our center for biopsy are not known, except for the PI-RADS score. Regarding the methodology of the bpMRI performed at our center for lesion segmentation and expert interpretation of PI-RADS v 2.1, these are defined in lines 104–114. There were no cases with a PI-RADS 1 score in this validation series.

  1. While if we look for AUC results they are on point, however comparison with other models or clinical standard output for better insight of the results would be welcome.

Response: Thank you. In a validation study, the discriminatory ability and clinical effectiveness of a model should be compared with those observed in the model's development cohort and other validations. We believe that comparing the calibration, ROC curves and DCAs with those obtained in the initial development and initial external validation would be valuable; therefore, we have included them in the new Figures 1, 2, and 3, referenced to the original publication.

  1. Emphasize sensitivity, specificity, and confidence intervals in Results much more and provide estimator-based results in Results instead of Interpretations and Discussion sections only to present the results more robustly.

Response: Thak you. The clinical utility of the BCN-MRI PM has been reinforced in the results, including confidence intervals, which have also been rewritten in the subsection 3.2, in lines 197-254.

  1. Strong clinical utility section, but a broader impact section discussing patient outcomes or cost-effectiveness might improve relevance.

Response: Thank you. The clinical utility of the BCN-MRI PM has been reinforced in the results, including confidence intervals, which have also been rewritten in the subsection 3.2, in lines 197-254. Regarding the cost-effectiveness of using the BCN-MRI PM we have introduced a sentence in the discussion section in lines 320-324.

  1. Transparency in experimental methods and results appears to be a strength of this study, however, reporting of the generalizability of the model results across differing patient populations may be required to improve the paper further.

Response: Thank you. A successful validation enables the use of a predictive model in the analyzed population. This means that the validation allows the BCN-MRI predictive model (PM) to be used in the population undergoing mapping biopsy of suspicious lesions, as mentioned in lines 266–280.

  1. Further explanation of BCN-MRI risk calculator usability and integration into clinical systems would be helpful.

Response: We have introduced some details according to the usability of the BCN-MRI risk calculator in lines 72-81.

  1. Providing graphical representations of the model performance, like ROC curves or calibration plots, could help minimize this problem.

Response: Thank you. We have incorporated a modification of Figure 1, 2 and 3, in which it is now incorporating the calibration plots, ROC curves and DCAs corresponding to the development cohort and initial external validation have been included, referenced to reference [10].

  1. Reduce the similarity, as it is copied from two papers, and the author has to make an effort to reduce the plagiarism.

Response: Thank you for the recommendation. We have reviewed the text and now plagiarism rate was 12% in the introduction, and 2% in the discussion.

  1. References can be cited that will help to better understand the broader applications of novel imaging techniques and predictive modelling in cancer detection and treatment field Papers like 
  1. Wang, H., Yan, Z., Yang, W., Liu, R., Fan, G., Gu, Z.,... Tang, Z. (2025). A strategy of monitoring acetylcholinesterase and screening of natural inhibitors from Uncaria for Alzheimer's disease therapy based on near-infrared fluorescence probe. Sensors and Actuators B: Chemical, 424, 136895. doi: https://doi.org/10.1016/j.snb.2024.136895
  1. He, Y., Bao, M., Chen, Y., Ye, H., Fan, J.,... Shi, G. (2024). Accuracy characterization of Shack–Hartmann sensor with residual error removal in spherical wavefront calibration. Light: Advanced Manufacturing, 4(4), 393-403. doi: 10.37188/lam.2023.036
  1. Xu, X., Luo, Q., Wang, J., Song, Y., Ye, H., Zhang, X.,... Shi, G. (2024). Large-field objective lens for multi-wavelength microscopy at mesoscale and submicron resolution. Opto-Electronic Advances, 7(6), 230212. doi: 10.29026/oea.2024.230212
  1. Zou, Y., Zhu, S., Kong, Y., Feng, C., Wang, R., Lei, L.,... Chen, L. (2024). Precision matters: the value of PET/CT and PET/MRI in the clinical management of cervical cancer. Strahlentherapie und Onkologie. doi: https://doi.org/10.1007/s00066-024-02294-8
  1. Yao, X., Zhu, Y., Huang, Z., Wang, Y., Cong, S., Wan, L.,... Hu, Z. (2024). Fusion of shallow and deep features from 18F-FDG PET/CT for predicting EGFR-sensitizing mutations in non-small cell lung cancer. Quantitative Imaging in Medicine and Surgery 2024, 14(8), 5460-5472. doi: 10.21037/qims-23-1028
  1. Sun, T., Lv, J., Zhao, X., Li, W., Zhang, Z.,... Nie, L. (2023). In vivo liver function reserve assessments in alcoholic liver disease by scalable photoacoustic imaging. Photoacoustics, 34, 100569. doi: https://doi.org/10.1016/j.pacs.2023.100569
  1. Du, Y., Chen, L., Yan, M., Wang, Y., Zhong, X., Xv, C.,... Cheng, Y. (2023). Neurometabolite levels in the brains of patients with autism spectrum disorders: A meta-analysis of proton magnetic resonance spectroscopy studies (N = 1501). Molecular Psychiatry, 28(7), 3092-3103. doi: 10.1038/s41380-023-02079-y
  1. Xiang, Y., Jialing, W., Jianhao, L., Jiangpeng, A., Feizhou, D.,... Rui, J. (2024). Bi-Parametric Magnetic Resonance Imaging Analysis of Biochemical Recurrence of Prostate Cancer after Radical Surgery and Its Predictive Value: A Retrospective Study. Archivos Españoles de Urología, 77(5), 598-604. doi: 10.56434/j.arch.esp.urol.20247705.81
  1. Lan, Z., Tan, F., He, J., Liu, J., Lu, M., Hu, Z.,... Huang, Y. (2024). Curcumin-primed olfactory mucosa-derived mesenchymal stem cells mitigate cerebral ischemia/reperfusion injury-induced neuronal PANoptosis by modulating microglial polarization. Phytomedicine, 129, 155635. doi: https://doi.org/10.1016/j.phymed.2024.155635
  1. Li, W., Wu, J., Zhang, J., Wang, J., Xiang, D., Luo, S.,... Liu, X. (2018). Puerarin-loaded PEG-PE micelles with enhanced anti-apoptotic effect and better pharmacokinetic profile. Drug Delivery, 25(1), 827-837. doi: 10.1080/10717544.2018.1455763
  1. Gao, X., Tang, J., Liu, H., Liu, L., & Liu, Y. (2019). Structure–activity study of fluorine or chlorine-substituted cinnamic acid derivatives with tertiary amine side chain in acetylcholinesterase and butyrylcholinesterase inhibition. Drug Development Research, 80(4), 438-445. doi: https://doi.org/10.1002/ddr.21515
  1. Lu, Q., Chen, Y., Liu, H., Yan, J., Cui, P., Zhang, Q.,... Liu, Y. (2020). Nitrogen-containing flavonoid and their analogs with diverse B-ring in acetylcholinesterase and butyrylcholinesterase inhibition. Drug Development Research, 81(8), 1037-1047. doi: https://doi.org/10.1002/ddr.21726
  1. Wang, Y., Xu, Y., Song, J., Liu, X., Liu, S., Yang, N.,... Zhang, Y. (2024). Tumor Cell-Targeting and Tumor Microenvironment–Responsive Nanoplatforms for the Multimodal Imaging-Guided Photodynamic/Photothermal/Chemodynamic Treatment of Cervical Cancer. International Journal of Nanomedicine, 19, 5837-5858. doi: 10.2147/IJN.S466042

Response: We recognize this comment, and we have added a last sentence in the discussion reflecting your suggestion (lines 331-333), also including your suggested references [34-47].

Comments on the Quality of English Language

The English could be improved to more clearly express the research.

Response: a new review of English has been made.

Round 2

Reviewer 1 Report

Comments and Suggestions for Authors

Overall, while the authors provide some explanations and justifications, their responses often fall short of fully addressing the comments and suggestions. Additional efforts to include a sensitivity analysis for PI-RADS 2 lesions, clarify the ethical safeguards in patient communication and consent, expand on the limitations of biopsy-based validation, and explore subgroup analyses are necessary to address these concerns comprehensively and improve the robustness and clinical relevance of the manuscript.

Author Response

Comment: Overall, while the authors provide some explanations and justifications, their responses often fall short of fully addressing the comments and suggestions. Additional efforts to include a sensitivity analysis for PI-RADS 2 lesions, clarify the ethical safeguards in patient communication and consent, expand on the limitations of biopsy-based validation, and explore subgroup analyses are necessary to address these concerns comprehensively and improve the robustness and clinical relevance of the manuscript.

Response: We appreciate the  Reviewer´s additional comments.  Below we address each of them and propose modifications in the manuscript, which will be highlighted in red.

  1. General Feedback: While the authors provide some explanations and justifications, their responses often fail to fully address the comments and suggestions.

Response: We fully agree with the Reviewer that individuals with PI-RADS 2 should not undergo biopsy, as there is a general consensus on this matter. However, as noted in the meta-analysis of Oerther regarding sPCa detection in PI-RADS v 2.1, up to 9% of these cases may present with sPCa (lines 255–257 and reference [25]). Therefore, biopsies are recommended only for cases with specific features suggesting the presence of significant PCa, particularly a PSA density greater than 0.15, suspicious digital rectal examination findings, or concerning PSA increases.

At our center, we perform transperineal semi-automated prostate biopsies upon referral by urologists based on clinical information and external mpMRI results. In our series, we identified seven men (1.5%) referred with an external PI-RADS 2 classification. As explained in the Materials and Methods section (lines 109–112), a bpMRI is always conducted at our center to segment suspicious lesions, with classification by an expert radiologist using PI-RADS version 2.1. This PI-RADS score, rather than the external score, is reported and utilized for the validation of the BCN-MRI PM, as we consider it to be the most accurate.

We did not report the external PI-RADS score in our manuscript. Clinical consequences of reclassifications from an expert reading of a new bpMRI performed for segmentations have already been detailed in a prior publication (Clinical Genitourinary Cancer, 2024, PubMed). Following reclassification of external PI-RADS (of unknown version), we identified 73 individuals who underwent biopsy with a PI-RADS 2 (v2.1) classification. Of these, only wo cases (2.7%) revealed sPCa.

  1. Additional Efforts for Sensitivity Analysis for PI-RADS 2 Lesions:

Response: Thank you. Further analyzing the potential clinical utility of the BCN-MRI PM in individuals with PI-RADS 2, the probability of significant PCa ranged from 0.22% to 26.70% among the 73 cases analyzed. The probabilities in the two cases with significant PCa (sPCa) were 11.28% and 22.99%, with respective PSA densities of 0.14 and 0.17, and normal digital rectal examinations in both. By applying an 11% threshold  for recommending biopsy in PI-RADS 2 cases, a 100% sensitivity for sPCa cases would be detected, while prostate biopsies using this threshold would be indicated in 16 cases (21.9%) within this subgroup.

  1. Ethical Considerations in Communication and Consent:

Response: Information about the prostate biopsy procedure and participating in this validation study was performed at our center during the initial medical visit, where the patient's characteristics and the urologist's referral request for prostate biopsy were reviewed. The informed consent forms were given to the individuals during this visit and must be signed and returned on the day of the procedure.

  1. Expanding the Limitations of Biopsy-Based Validation:

Response: In the revised version of the manuscript, the limitations of our study have been expanded (lines 328–347).

  1. Subgroup Analyses to Address Concerns and Enhance Clinical Relevance:

Response: Thank you for raising this concern. We believe the most relevant approach is to report the utility of the BCN-MRI PM according to PI-RADS scores. Based on this subanalysis, as detailed above for PI-RADS 2, we find that for PI-RADS 4 and 5, no threshold allows avoiding biopsies without missing sPCa. For PI-RADS 3 (the most uncertain scenario), 35 cases of sPCa were identified among 106 individuals. By applying a 10% threshold, 36 prostate biopsies (40%) could be avoided while 8 sPCa cases (22.8%) would be missed, representing a global loss of 3% among the 267 significant PCa cases diagnosed in the overall series.

In summary, the BCN-MRI PM shows potential utility in detecting sPCa among individuals classified as PI-RADS 2 and 3 (v2.1). While this analysis deviates from the predictive model validation focus of our study, such investigations have already been conducted during the initial development and external validation of the BCN-MRI PM (lines 79–80 [10, 11]) and in comparison, with the Rotterdam-MRI PM (lines 80–81 [12]).

Although reporting this analysis was not the primary objective of our study, we have added a subsection in Results entitle “3.4. Potential clinical utility of the BCN-MRI PM according to the PI-RADS v 2.1 score”, reporting this results in lines 250-263. An additional comment  has been also added in lines 319-327 at the discussion section, as well as another comment in the limitations paragraph in lines 342-347.

Reviewer 2 Report

Comments and Suggestions for Authors

Please revise the title to make it more concise and specific.

Proofreading is required.

Please provide the uploaded file for the correct one, as there is an inconsistency in the reference list. There are 45 citations in the text, but only 35 items in the bibliography. Plus, you're still at 39% for the plagiarism match. I will check the final version before accepting the paper.

Comments on the Quality of English Language

 The English could be improved to more clearly express the research.

Author Response

We appreciate comments and suggestions and  the possibility to improve the manuscript. Changes in the manuscript are highlighted in red.

  1. Please revise the title to make it more concise and specific.

Response: The title has been changed by one more concise and specific by adding significant prostate cancer detection in mapping 0.5 mm-core targeted biopsies.

  1. Proofreading is required.

Response: a new proofreading has been made

  1. Please provide the uploaded file for the correct one, as there is an inconsistency in the reference list. There are 45 citations in the text, but only 35 items in the bibliography. Plus, you're still at 39% for the plagiarism match. I will check the final version before accepting the paper.

Response: Sorry for these inconveniences. The current number of citations in the text is 47 corresponding to the 47 items in the bibliography. The manuscript has been rewritten for avoiding plagiarisms, especially in the section of material and methods.

  1. Comments on the Quality of English Language

The English could be improved to more clearly express the research.

Response: A new proof reading has been made

Round 3

Reviewer 2 Report

Comments and Suggestions for Authors

The authors have addressed all the comments.

Comments on the Quality of English Language

The English could be improved to express the research more clearly.